# Drug Delivery Systems Based on Pluronic Micelles with Antimicrobial Activity

**DOI:** 10.3390/polym14153007

**Published:** 2022-07-25

**Authors:** Corina Popovici, Marcel Popa, Valeriu Sunel, Leonard Ionut Atanase, Daniela Luminita Ichim

**Affiliations:** 1“Cristofor Simionescu” Faculty of Chemical Engineering and Environmental Protection, “Gheorghe Asachi” Technical University of Iasi, 700050 Iasi, Romania; corinapopovici77@yahoo.com; 2Academy of Romanian Scientists, 050045 Bucharest, Romania; 3Faculty of Chemistry, “Alexandru Ioan Cuza” University of Iasi, 700506 Iasi, Romania; vsunel@uaic.ro; 4Faculty of Medical Dentistry, “Apollonia” University of Iasi, 700511 Iasi, Romania; danielaluminitaichim@yahoo.com

**Keywords:** Pluronic, F127, polymeric micelles, drug delivery system, active pharmaceutical ingredient, antimicrobial activity

## Abstract

Bacterial oral diseases are chronic, and, therefore, require appropriate treatment, which involves various forms of administration and dosing of the drug. However, multimicrobial resistance is an increasing issue, which affects the global health system. In the present study, a commercial amphiphilic copolymer, Pluronic F127, was used for the encapsulation of 1-(5′-nitrobenzimidazole-2′-yl-sulphonyl-acetyl)-4-aryl-thiosemicarbazide, which is an original active pharmaceutical ingredient (API) previously synthesized and characterized by our group, at different copolymer/API weight ratios. The obtained micellar systems, with sizes around 20 nm, were stable during 30 days of storage at 4 °C, without a major increase of the Z-average sizes. As expected, the drug encapsulation and loading efficiencies varied with the copolymer/API ratio, the highest values of 84.8 and 11.1%, respectively being determined for the F127/API = 10/1 ratio. Moreover, in vitro biological tests have demonstrated that the obtained polymeric micelles (PMs) are both hemocompatible and cytocompatible. Furthermore, enhanced inhibition zones of 36 and 20 mm were observed for the sample F127/API = 2/1 against *S. aureus* and *E. coli*, respectively. Based on these encouraging results, it can be admitted that these micellar systems can be an efficient alternative for the treatment of bacterial oral diseases, being suitable either by injection or by a topical administration.

## 1. Introduction

Polymeric micelles (PMs) represent a distinct class of carriers formed from amphiphilic block or graft copolymers [1,2]. The driving force of the self-assembly of these amphiphilic copolymers in aqueous media, at the nanometric scale, is the thermodynamic incompatibility between the different sequences (hydrophilic/hydrophobic) [3]. During the last two decades, PMs have attracted a great interest, in particular owing to their strong ability to solubilize water-insoluble active principles and thus to be effective drug nanocarriers [4,5,6,7]. Like surfactant micelles, polymer micelles have a core–corona structure capable of encapsulation various types of molecules. In addition, PMs show greater resistance to the effects of dilution than surfactant micelles due to their generally very low CMC (in the order of mg/L). Moreover, it is possible to optimize their colloidal characteristics by modulating the nature of the hydrophobic or hydrophilic copolymers sequences. Thus, the structure of the hydrophobic core can be adjusted in order to optimize its affinity for the incorporated active ingredient. Finally, due to their small size and their hydrophilic surface, PMs are poorly recognized by the immune system, which would make it possible to prolong the plasma half-life of their cargo. PMs therefore possess unique characteristics among all drug delivery systems [8,9,10].

Among the amphiphilic copolymers, one of the most studied systems are non-ionic poly(ethylene oxide)-poly(propylene oxide)-poly(ethylene oxide) (PEO-PPO-PEO) triblock copolymers, also called poloxamers or Pluronics^®^, which is a registered trademark of BASF [11]. In aqueous solution, these copolymers self-assemble to form spherical or elongated micelles, with a PPO core and PEO corona, but other structures may exist depending on the lengths of each block. F127, having a linear structure PEO_97_-PPO_69_-PEO_97_ (indexes are polymerization degree values), is one of the most suitable Pluronics for preparation of drug delivery systems as it has a hydrophilic/lipophilic balance (HLB) of 22, a molecular weight of 12,600 g/mol, low critical micellar concentration (CMC = 0.0031%, *w*/*w*), excellent biocompatibility and satisfactory safety [12,13]. In addition, owing to their small CMC, F127 micelles are stable to dilution in the presence of biological fluids.

Different types of active principles can be solubilized within the PPO hydrophobic core [14] or can be conjugated to the PEO corona [15]. The loading of hydrophobic molecules in this type of PMs, with sizes usually smaller than 100 nm, can increase their solubility, stability and improve their pharmacokinetics as well as biodistribution [14].

One of the most important advantage of these copolymers is that their hydrophilic PEO corona prevents aggregation, protein adsorption and especially recognition by the reticuloendothelial system (RES), leading thus to an increased blood circulation time. Due to these important advantages, the first micellar formulation approved for clinical trials was based on doxorubicin-loaded Pluronic F127 and L61 [16].

A growing problem of modern society is related to the uncontrolled use of antibiotics which have as a consequence increased multimicrobial resistance, affecting thus the global health system. An extremely sophisticated microbial environment is located in the oral cavity, in which can be found around 700 strains that can contribute to the rapid development of oral diseases [17]. Even if the synthesis of new drugs with antimicrobial activity has recorded remarkable progress in the last few years, their low solubility and bioavailability, high toxicity and inadequate release profile limits their clinical utilization [18]. In order to overcome these drawbacks, different types of PMs were studied for the preparation of formulation with antimicrobial activity [19,20,21,22,23]. Among those, we are interested in the use of Pluronics as nanocarriers for active principles with antimicrobial activity.

Hashemi et al. [24] encapsulated antimicrobial peptides in P127 micelles and observed, by ex vivo tests, that the obtained micellar formulation reduced the populations of fungal pathogens in tracheal and lung tissue.

In another study, Purro et at. [25] obtained a conjugate between Pluronic P127 and the siderophore desferrioxamine B (DFO) complexed to Ga, DFO:Ga^III^ (DG). These authors demonstrated that their conjugate was efficient against strains of *P. aeruginosa*. 

Recently, it was proved that the inclusion of photosensitizers, based on cationic porphyrins, into the Pluronic polymer micelles (F-127) significantly increased the efficiency of the antimicrobial photodynamic inactivation [26].

Based on the evident advantages of the Pluronic micelles, we propose in this study the preparation and characterization of a micellar system with enhanced antimicrobial activity due to an encapsulated original active pharmaceutical ingredient (API), 1-(5′-nitrobenzimidazole-2′-yl-sulphonyl-acetyl)-4-aryl-thiosemicarbazide, which was previously synthesized [27]. The chemical structure of this active principle is provided in Figure 1.

The obtained micellar system was both physicochemical and biological characterized. Moreover, the antimicrobial activity was assessed on different Gram-positive and Gram-negative bacteria.

## 2. Experimental Section

### 2.1. Materials

The active pharmaceutical ingredient 1-(5′-nitrobenzimidazole-2′-yl-sulphonyl-acetyl)-4-aryl-thiosemicarbazide was synthesized and fully characterized in a previous publication [27]. Pluronic F127 was purchased from Sigma Aldrich (Steinheim, Germany). All the solvents were used as received and without further purification. Human dermal fibroblasts cell line (HDFa) and the necessary supplies (antibiotic cocktail: penicillin and streptomycin, non-essential amino acids, trypsin solution and fetal bovine serum (FBS)) for in vitro cytotoxicity assay were purchased from Thermo Fisher Scientific (Waltham, MA, USA). Freeze-dried stains (*Escherichia coli*-ATCC^R^ 11775^TM^, *Pseudomonas aeruginosa*-ATCC^R^ 10154^TM^, *Klebsiella pneumoniae*-ATCC^R^ BAA-1705^TM^, *Staphylococcus aureus*-ATCC^R^ 25 923^TM^ and *Porphyromonas gingivalis*-ATCC^R^ 33277^TM^) were purchased from ATCC (Manassas, VA, USA). Chapman agar (mannitol salt agar) was purchased from Oxoid (Hampshire, UK) and MacConkey agar from G&M Procter Ltd. (Perth, UK). 

### 2.2. Methods

#### 2.2.1. Micelle’s Preparation Procedure

Preparation of PMs was carried out by a dialysis method starting from a common solvent. In a typical procedure, 100 mg of Pluronic F127 was added in 10 mL of dimethylsulfoxide (DMSO) solution and stirred at room temperature until complete dissolution of copolymer. Afterwards, the solution was dialyzed against 1 L of ultrapure water using cellulose dialysis membranes (molecular weight cut off: 12 kDa, manufacturer, Sigma Aldrich, Steinheim, Germany). The water was changed eight times during 24 h of dialysis. The dry powder was collected after the freeze drying of the micellar solutions and then was stored at −4° before further use.

A similar procedure was used for the preparation of API-loaded PMs with the difference that at the solution of the block copolymer in DMSO were added different amounts of API in order to have three weight ratios between the copolymer and the active principle, such as: 10:1; 5:1 and 2:1.

#### 2.2.2. Physicochemical Characterization Methods

The micellar sizes were investigated by Dynamic Light Scattering (DLS) measurements in phosphate buffer solution (PBS; pH = 7.4), this medium being similar to the in vivo medium. DLS were carried out on a Malvern Zetasizer Pro (Malvern Pananalytical, Worcestershire, UK), using the NIBS (Non-Invasive BackScattering) technology, equipped with a 4 mW He–Ne laser operating at a wavelength of 532 nm and at a scattering angle of 173°. The software package of the instrument calculates, by using the Stokes–Einstein equation, the hydrodynamic diameter (volume average) Dv, the Z-average diameter, which is an intensity-weighted size average and the polydispersity index (PDI) of the sample. In order to determine the mean diameter of the particles, the data were collected in automatic mode, typically requiring a measurement duration of 70 s. For each experiment, five consecutive measurements were carried out. The stability of the micellar solutions, stored at 4 °C, was assessed as a function of time for 30 days. The zeta potential values of PMs were determined by electrophoresis in phosphate buffer solution (PBS; pH = 7.4) using the same instrument. 

In order to determine the encapsulation efficiency of API, calibration curves were constructed in DMSO, using different concentrations of API, and their absorbance values were recorded on a Nanodrop spectrophotometer (Thermo Scientific, Waltham, MA, USA) at the wavelength of 480 nm. A known amount of the API-loaded micelles, as powder, was dissolved in 1 mL of DMSO in order to completely destroy the micelles and to release the loaded active principle. The amount of API from the micelles was spectrophotometrically quantified, based on the calibration curve, using a UV spectrometer (Nanodrop One, Thermo Scientific, Waltham, MA, USA). Drug encapsulation efficiency (DEE) and drug loading efficiency (DLE) were calculated using Equations (1) and (2), respectively:(1)$$\mathrm{DEE}\left(\%\right)=\frac{amount\;of\;drug\;in\;micelles}{amount\;of\;added\;drug}\times 100$$
(2)$$\mathrm{DLE}\left(\%\right)=\frac{amount\;of\;drug\;in\;micelles}{amount\;of\;added\;polymer\;and\;drug}\times 100$$

Three determinations were performed for each sample and the errors were ±0.3%.

#### 2.2.3. Biological Characterization Methods

The hemolytic potential of the obtained PMs was evaluated using a spectrophotometric method adapted from Rata et al. [28]. These tests were started after obtaining the institutional ethical authorization (28/02.06.2022-The scientific research ethics committee) and the appropriate informed consent. The blood from the healthy non-smoking human volunteer was collected in vacutainer tubes and treated with PMs. In total, 5 mL anti-coagulated blood was centrifuged at 2000 rpm (RCF = 381× g) for 5 min and washed with normal saline solution several times to completely remove the plasma and obtain erythrocytes. After purification, erythrocytes were re-suspended in 25 mL normal saline solution. PMs saline solution with different concentrations (0.5 mL) was added to 0.5 mL of erythrocytes suspension (final concentrations were 10, 50, 100, and 200 mg PMs/mL erythrocytes suspension). Positive (100% lysis) and negative (0% lysis) control samples were prepared by adding equal volumes (0.5 mL) of Triton X-100 and a standard saline solution. The samples were incubated at 37 °C for 180 min. Once every 30 min the samples were gently shaken to re-suspend erythrocytes and PMs. After the incubation time, the samples were centrifuged at 2000 rpm (RCF = 381× *g*) for 5 min and 100 µL of supernatant was incubated for 30 min at room temperature to allow hemoglobin oxidation. Oxyhemoglobin absorbance in supernatants was measured at 540 nm using a Nanodrop One UV-Vis Spectrophotometer from Thermo Fischer Scientific, Waltham, MA, USA. All samples were analyzed in triplicate. The hemolytic percentage was calculated using Equation (3):(3)$$\mathrm{Haemolysis}\;\left(\%\right)=\frac{\left(A_{S}-A_{NC}\right)}{\left(A_{PC}-A_{NC}\right)}\times 100$$
where, *A_s_* is the absorbance of the sample; *A_NC_* and *A_PC_* are the absorbance values of the negative and positive control, respectively.

The MTT method was applied to assess the in vitro cytotoxicity of free and API-loaded PMs by using adherent adult human fibroblast cells of dermal origin (HDFa). After thawing the fibroblast cells in the thermostatic bath at 37 °C (Digital thermostatic baths, DIGIBATH 2–BAD\2RAYPA, Spain) the cells (HDFa) were cultured in complete medium grow: DMEM (Dulbecco’s Modified Eagle Medium) supplemented with 10% fetal bovine serum, 1% antibiotics and 1% non-essential amino acids, at 37 °C and a humidified atmosphere of 5% CO_2_ (MCO-5AC CO_2_ Incubator, Panasonic Healthcare Co., Ltd., Sakata Oizumi-Machi Ora-Gun Gunma, Japan). Cells were allowed to proliferate in culture flasks (NuncTM EasYFlask 25 cm^2^ TM, ThermoFisher Scientific, Roskilde, Denmark) to reach 80% confluence, then were trypsinized with 0.05% trypsin solution at 37 °C, followed by the addition of complete medium to neutralize the trypsin, were centrifuged (Rotofix-32A, Hettich, Andreas GmbH Hettich & Co.KG, Tuttlingen, Germany) and re-suspended in complete medium DMEM. For performing the in vitro cytotoxicity assay, the reagents were purchased from Thermo Fisher Scientific. After centrifugation and re-suspension in fresh medium, the viable cells were plated in on flat-bottom 96-well plates (TPP Techno Plastic Products AG, Switzerland) and incubated for 24 h. After 24 h of incubation the culture grow medium was replaced with fresh medium, and the test material were put in direct contact with the fibroblast cells at three types of concentrations: 10, 50 and 100 µg/mL. Prior to performing the cytotoxicity test, the materials were sterilized with UV-VIS radiation for 3 min. Microscopic analysis was performed on the inverted optical microscope (CKX41, Olympus, Tokyo, Japan) with a built-in camera and QuickPHOTO camera 3.0 software. Cell viability determination was performed at 24 and 48 h after incubation of the cells treated by the quantitative colorimetric assay with tetrazolium salt (3-(4,5-dimethylthiazol-2-yl)-2,5-diphenyltetrazolium bromide (MTT)), (Merck Millipore, Darmstadt, Germany). After 24 h incubation, 100 μL of culture medium was replaced with 100 μL of fresh medium followed by adding 10 μL of MTT dye to each well and incubating for 4 h at 37 °C with 5% CO_2_. After 4 h of incubation, 90 μL of medium was removed from each well and 100 μL of DMSO was added to dissolve the formazan crystals, followed by re-incubation for 10 min at 37 °C. The absorbance was measured at 570 nm using a Multiskan FC automatic plate reader (Thermo Fisher Scientific Oy, Finland) with Sknalt Software 4.1 software. Each sample was tested in triplicate, and cell viability was expressed as % of untreated cells (control) considered 100% viable.

Testing on the antimicrobial effect of the API, before and after incorporation into PMs based on F127, was performed by determining the diameter of the zone of inhibition by diffusion method, using three reference strains of Gram-negative bacteria (*Escherichia coli* (ATCC^R^ 11775^TM^), *Pseudomonas aeruginosa* (ATCC^R^ 10154^TM^), *Klebsiella pneumoniae* (ATCC^R^ BAA—1705^TM^)), a Gram-pozitive bacteria (*Staphylococcus aureus* (ATCC^R^ 25 923^TM^)), and an anaerobic bacterium (*Porphyromonas gingivalis* (ATCC^R^ 33277^TM^)). The inoculum used in the five test microorganisms used for seeding had a turbidity of 0.5 McFarland. On each plate of culture medium, in the wells made, 100 µL of the micellar suspension with a concentration of 200 µg/mL was placed. The diameters of the inhibition zones were measured after incubation at 37 °C, for 24 h in the case of *Staphylococcus aureus*, *Escherichia coli*, *Pseudomonas earuginosa*, *Klebsiella pneumoniae* and 48 h for *Porphyromonas gingivalis* (grown in Anaeroar, under anaerobic conditions). It was considered that the tested samples have an antimicrobial activity if around the wells there were areas of inhibition of the growth of the test microorganism, with a significantly large diameter, i.e., at least 15 mm.

The experiments were repeated three times, and the results (mm area of inhibition) were expressed as mean values.

## 3. Results

### 3.1. Micellar Sizes and Stability

Generally, in order to have the highest drug-loading efficiency during its circulation in blood, the micelles should be small enough to evade detection and destruction by the reticular endothelial system. The obtained micellar system loaded with API was designed in order to be administrated either by injection or by a topical application. Independent of the administration route, an important characteristic of this type of drug delivery systems is represented by their size and polydispersity index. In Table 1 are presented the colloidal characteristics of the free and API-loaded PMs.

From this table it is possible to notice that the micellar Dv of free micelles is around 20 nm, which is in concordance with the literature data concerning the F127 micelles [29]. It can also be observed that only a small increase of the Dv is noticed when the API is loaded at different copolymer/API ratios. This behavior can be explained be a modification of the aggregation number by the fact that the encapsulation of a hydrophobic molecule, as it is the case for our active principle, leads to a more compact micellar core. In Figure 2 are shown the size distribution curves in volume for free and API-loaded PMs at 37 °C. 

From Figure 2 it appears that these curves have a monomodal distribution and that the PDI does not increase drastically with increasing the amount of API loaded which indicates the fact that there are no micellar aggregates. However, from data provided in Table 1, it can be observed that the PDI values increase slightly with the increase of the copolymer/API ratio but these values are always under 0.3. Moreover, from this table it appears that the ZP values are almost neutral, as expected for non-ionic copolymers such as Pluronics [30,31]. In this case, the micelles are sterically stabilized by the PEO sequences.

Among the colloidal properties which are important characteristic of a colloidal solution is the suspension stability as a function of time. Figure 3 shows the evolution of the Z-average as a function of time for all prepared micellar suspensions.

Figure 3 illustrates that only a slight increase of the Z-average values is noticed after 20 days for the sample F127/API = 2/1. For the other analyzed samples, the micellar suspension is stable, within the experimental error limits, during 30 days which, is an advantage for the long term usage and storage of these systems.

### 3.2. Drug Encapsulation Efficiencies

The drug encapsulation efficiency (DEE) and drug loading efficiency (DLE) of different samples are provided in Table 2.

From Table 2, it appears that API was successfully incorporated into the PMs and that both the DEE and DLE values are increased by decreasing the initial copolymer/API ratio. At a F127/API ratio of 10/1, the DEE has the highest value of 84.8%, whereas the DLE is equal to 11.1%. These values are comparable with other values in the literature concerning the encapsulation of other active molecules in Pluronics micelles [31].

The main aim of this study was the preparation of a micellar system loaded with an active principle having an antimicrobial activity and which could be used for the treatment of oral diseases. As it was demonstrated that the API-loaded PMs are stable in time, it was of interest to study also the release kinetics of the API in PBS (pH = 7.4) at 37 °C. In Figure 4 are given the curves of the cumulative release kinetics for free API and also for the API-loaded PMs.

A first observation from Figure 4 is related to the fact that the release kinetics of free API is higher than that of the encapsulated active principle. This behavior is explained by the fact that the API loaded within the micelles must pass through a barrier formed by the copolymer core and corona. Comparing the micellar systems with loaded API, it appears that the release kinetics are higher for the sample F127/API = 10/1, where the amount of API is the lowest. Moreover, the release kinetic curves are typical for a release controlled by diffusion in which the equilibrium is reached after 24 h even if not all the amount of API was released. 

### 3.3. Assessment of the Haemolysis Degree

Hemolysis is the destruction of red blood cells along with the release of hemoglobin and other internal components into the surrounding fluid. If this destruction occurs in a significant number of red blood cells in the body, it can lead to dangerous pathological conditions [32,33]. Therefore, all biomedical products designed for intravenous administration should be evaluated for their hemolytic potential [34]. The small sizes of this micellar system make it suitable for administration by injection. In this case, it was worthy to study their interaction with the blood components. Figure 5 shows the evolution of the hemolysis degree as a function of time and concentration.

It is well known that a material is considered as being hemotoxic if the hemolysis degree is higher than 5% [35]. From Figure 5 it can be noticed that even at the highest concentration of 200 μg/mL, the hemolysis degree is smaller than 3% which proves that our micellar systems are hemocompatible.

### 3.4. In Vitro Cytotoxicity Analysis

As these micellar systems are designed for biomedical applications, it was of interest to study their cytotoxicity in order to assess their biocompatibility. Cellular viability was evaluated on human fibroblast after 24 and 48 h, and the results are given in Figure 6.

The data provided in Figure 6 show, first of all, that the tested PMs at all four established concentrations (10, 50, 100 and 200 µg/mL) did not show cytotoxic effects on fibroblast cells at both 24 and 48 h after incubation. Then, it can be observed that the cellular viability decreases with increasing the concentration, but, even at the highest concentration of 200 μg/mL and 48 h, the cellular viability remains higher than 80%, which is the threshold from which a material is considered to be cytotoxic. This biocompatible behavior is also illustrated by the photographs given in Table 3.

The photographs given in Table 3 are supplementary proof that the prepared micellar system has a high cellular viability, and, therefore, it can be considered as safe for biomedical applications.

### 3.5. In Vitro Antimicrobial Activity

Finally, owing to the chemical structure of our original active principle, it was of interest to investigate the antimicrobial activity of the obtained micellar systems in order to determine if they can be potentially used for the treatment of oral diseases (especially the bacterial plaque). Different bacterial strains were selected based on their presence in the oral cavity and the obtained results are summarized in Table 4. 

From this table, it is obvious that free micelles have the lowest antimicrobial activity. In the case of free API, it clearly appears that it has an enhanced antimicrobial activity on *E. coli* (25 ± 0.4 mm) and especially on *S. aureus* (39 ± 0.6 mm) with inhibition diameters much higher than 15 mm. In addition, a medium activity was observed for *K. Pneumonias* (14 ± 0.5 mm), whereas almost no activity was detected against *P. aeruginosa* and *P. gingivalis*. This tendency is also confirmed for API-loaded PMs. In fact, by increasing the F127/API ratio, the antimicrobial activity increases, approaching very much the value of free API. Our results suggest that the antibacterial activity is higher against Gram-positive bacteria than Gram-negative ones. A Gram-positive bacillus does not have an outer cell wall beyond the peptidoglycan membrane, which makes it more absorbent for different molecules [36]. On the contrary, Gram-negative bacteria have a thin peptidoglycan layer and have an outer lipid membrane. This can contribute to the antibiotic resistance of the bacterial cells and be involved in the interaction with host cells. The difference in the antibacterial activity against the studied bacterial strains can be correlated with the structural difference between the lipid layer of these Gram-negative bacteria, which affects the cell permeability. Moreover, the lack of the antibacterial activity against *P. aeruginosa* and *P. gingivalis* demonstrate that our system is highly selective in killing different strains of bacteria, and this can be considered an important advantage. Concerning the molecular mechanism of the antibacterial activity of these types of compounds, it is still under debate in the literature, but it seems that the inhibitory action towards bacterial topoisomerases could be the main explanation [37,38]. Undoubtedly, more in-depth research in this area is urgently needed in order to establish the exact mechanism of action of these compounds. The photographs from Figure 7 are a good indication of this antimicrobial activity.

## 4. Conclusions

In this paper it was demonstrated that the polymeric micelles (PMs) obtained using a commercial available copolymer, Pluronic F127, can be loaded with an active pharmaceutical ingredient (API) at different copolymer/API weight ratios. The free and API-loaded PMs were characterized, by DLS, in terms of their size, stability and zeta potential. It was found that the loading of API in the free F127 micelles, of around 20 nm in diameter, has no significant impact either on the micellar size or on the polydispersity index. The API-loaded PMs show high stability over time, proven by almost constant average sizes over 30 days. In vitro tests have proven both their non-cytotoxic behavior, with cellular viabilities higher than 80%, and low degree of hemolysis (smaller than 3% even at high concentrations such as 200 μg/mL). The in vitro testing was completed by antimicrobial tests on several types of bacterial strains (Gram-positive and Gram-negative bacteria). An enhanced antimicrobial activity was observed on *S. aureus* and *E. coli*, whereas for *K. Pneumonias* only a medium one was recorded. Multidrug-resistant bacterial infection is expected to be overcome by these effective nanoscale drug carriers. Nanoscale drug PMs have been shown to be able to release the API in a controlled manner, increase the penetration to biofilms, improve drug stability and enhance drug bioavailability; therefore, it can be admitted that these PMs might be recommended as safe systems for further in vivo tests.

## Figures and Tables

**Figure 1 polymers-14-03007-f001:**
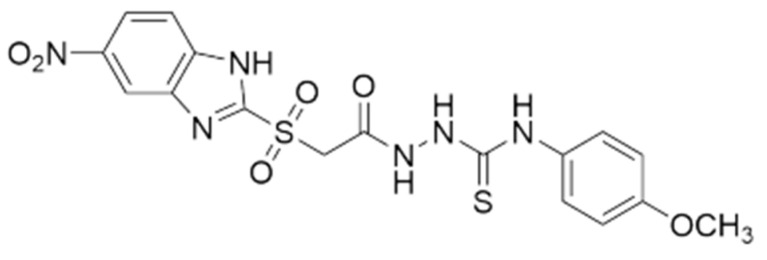
Chemical structure of the active pharmaceutical ingredient (API): 1-(5′-nitrobenzimidazole-2′-yl-sulphonyl-acetyl)-4-aryl-thiosemicarbazide.

**Figure 2 polymers-14-03007-f002:**
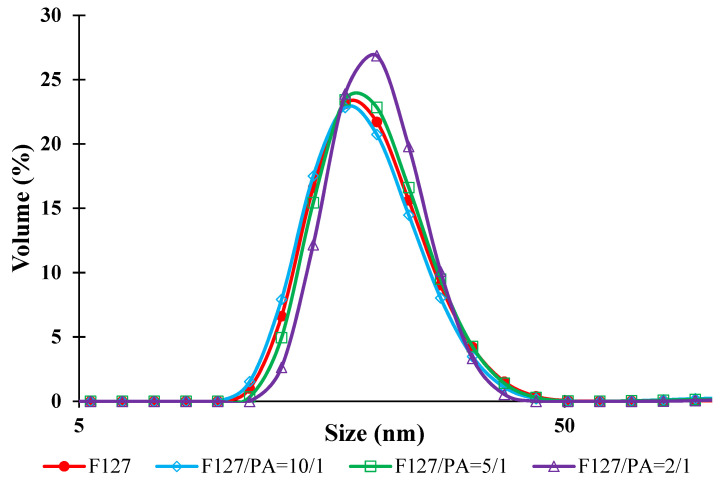
Size distribution curves in volume for free and API-loaded PMs at 37 °C.

**Figure 3 polymers-14-03007-f003:**
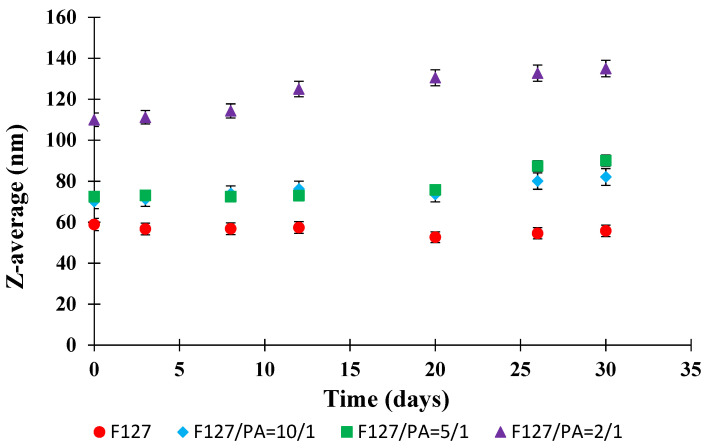
Evolution of the Z-average values as a function of time.

**Figure 4 polymers-14-03007-f004:**
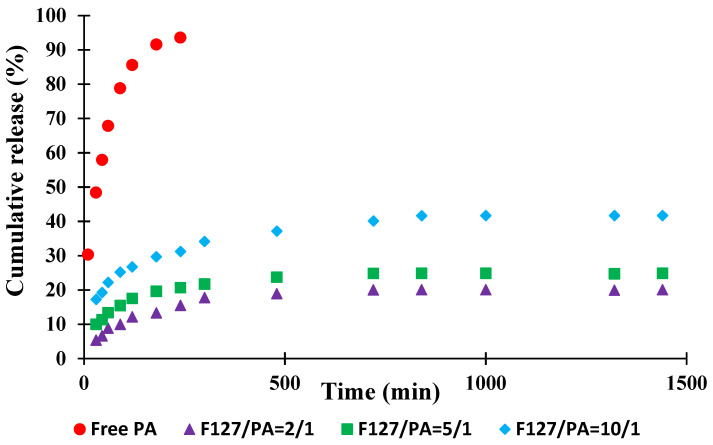
Evolution of the cumulative release kinetics in PBS (pH = 7.4) at 37 °C.

**Figure 5 polymers-14-03007-f005:**
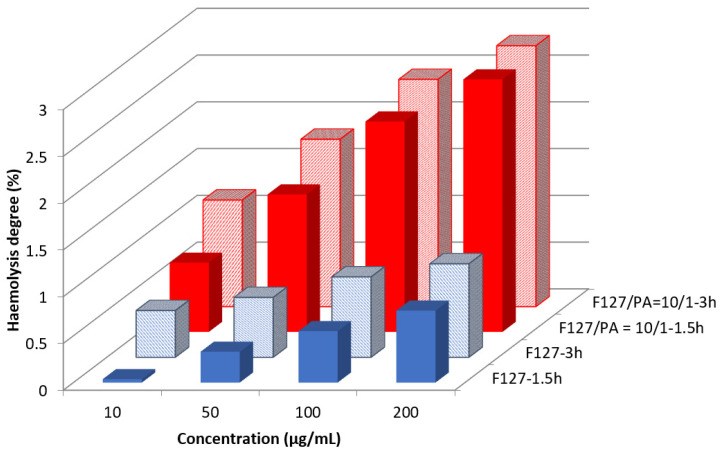
Evolution of the hemolysis degree for free F127 micelles and F127/API = 10/1 sample as a function of concentration and time.

**Figure 6 polymers-14-03007-f006:**
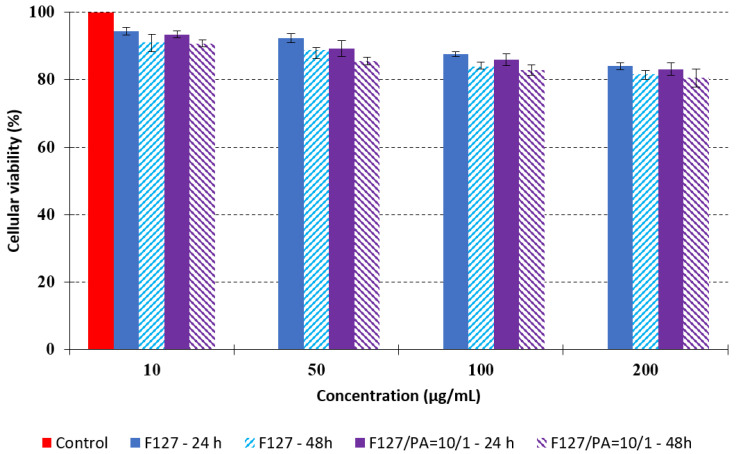
Evolution of the in vitro cellular viability after 24 and 48 h as a function of the concentration for sample F127/API = 10/1.

**Figure 7 polymers-14-03007-f007:**
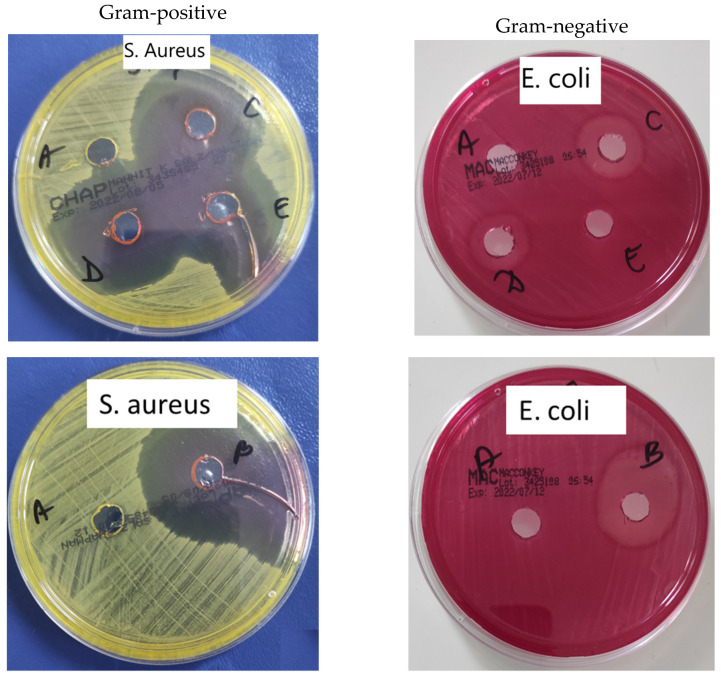
Antimicrobial activity of free and API-loaded PMs against Gram-positive and Gram-negative strains. A-negative control (free F127 micelles), B-positive control (API), C—F127/API = 2/1, D—F127/API = 5/1 and E—F127/API = 10/1.

**Table 1 polymers-14-03007-t001:** Diameter in volume (Dv), polydispersity index (PDI) and zeta potential (ZP) values of free and API-loaded PMs.

Sample	Dv (nm)	PDI	ZP (mV)
F127	20.3 ± 0.1	0.141	−3.0
F127/API= 10/1 (% *w*/*w*)	20.7 ± 0.2	0.198	−3.0
F127/API= 5/1 (% *w*/*w*)	20.3 ± 0.2	0.284	−2.8
F127/API = 2/1 (% *w*/*w*)	21.4 ± 0.3	0.308	−2.7

**Table 2 polymers-14-03007-t002:** DEE and DLE values for API-loaded PMs.

F127/API Ratio (% *w*/*w*)	DEE (%)	DLE (%)
10/1	84.8	11.1
5/1	66.6	10.3
2/1	30.9	7.7

**Table 3 polymers-14-03007-t003:** Micrographs of the fibroblast cells after 24 and 48 h.

Sample		Micrographs	
		24 h	48 h
Control		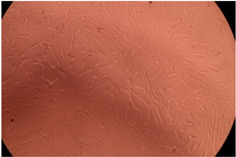	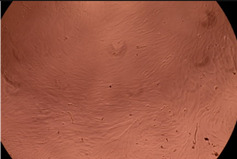
F127	10 µg/mL	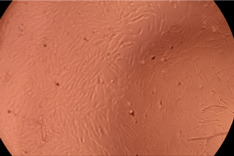	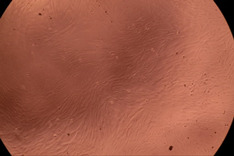
	50 µg/mL	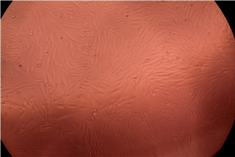	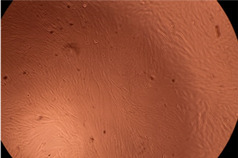
	100 µg/mL	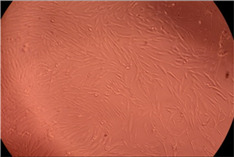	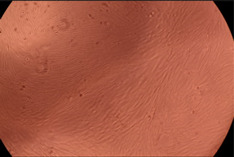
	200 µg/mL	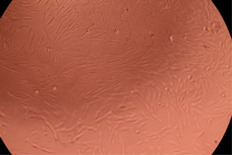	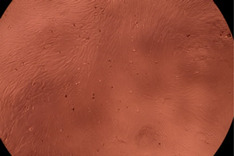
F127/PA = 10/1 (g/g)	10 µg/mL	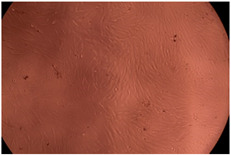	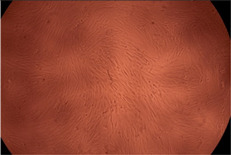
	50 µg/mL	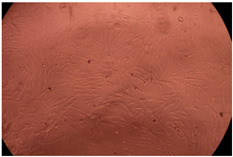	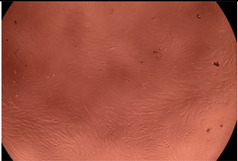
	100 µg/mL	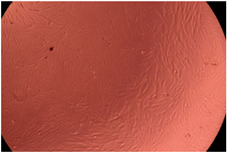	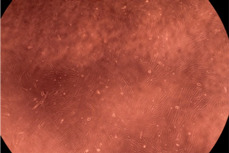
	200 µg/mL	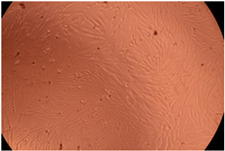	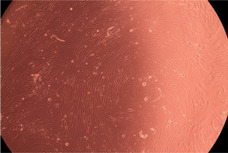

**Table 4 polymers-14-03007-t004:** Inhibition zone diameter (mm) for different bacterial strains in the presence of free PMs, free API and API-loaded PMs. Concentration was equal to 200 μg/mL.

Strain	Negative Control (Free PMs)	Positive Control (Free API)	F127/PA = 2/1	F127/PA = 5/1	F127/PA = 10/1
*Escherichia coli*	8 ± 0.2	25 ± 0.4	20 ± 0.4	17 ± 0.3	9 ± 0.4
*Pseudomonas aeruginosa*	8 ± 0.3	9 ± 0.3	9 ± 0.3	8 ± 0.2	8 ± 0.2
*Klebsiella pneumonia*	8 ± 0.2	14 ± 0.5	10 ± 0.4	9 ± 0.3	8 ± 0.3
*Staphylococcus aureus*	8 ± 0.2	39 ± 0.6	36 ± 0.6	35 ± 0.5	32 ± 0.5
*Porphyromonas gingivalis*	8 ± 0.3	9 ± 0.3	9 ± 0.3	8 ± 0.3	8 ± 0.3

## Data Availability

The data presented in this study are available on request from the corresponding author.

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
