# Peer review of "Drug Delivery Systems Based on Pluronic Micelles with Antimicrobial Activity"

_polymers, 2022, doi:10.3390/polym14153007_

Round 1

Reviewer 1 Report

Drug delivery systems based on pluronic micelles have been developed in this manuscript for the potential antimicrobial application. the active pharmaceutical ingredient [1-(5′-nitrobenzimidazole-2′- 94 yl-sulphonyl-acetyl)-4-aryl-thiosemicarbazide] was used as model for study the drug delivery properties of F127, which is a surfactant and have been widely used. The micellar sizes, stability, drug encapsulation efficiencies, haemolysis degree, In vitro cytotoxicity, and antimicrobial activity have been studied. An enhanced antimicrobial activity was observed on S. aureus and E. coli, whereas for K. Pneumonias only a medium one was recorded. In total, this is a well-organized manuscript, it could be accepted after addressing these issues:

1.     Authors must provide the SEM images to show the morphology of drug carrier.

2.     The control group of antimicrobial activity also need to be provided.

3.     Why there are almost no activity against P. aeruginosa and P. gingivalis? Why this happened, and the detailed antibacterial mechanism may be provided.

4.     Some related research about the antibacterial polymers should be cited to highlight the potential applications of these materials. Engineering of hollow polymeric nanosphere-supported imidazolium-based ionic liquids with enhanced antimicrobial activities. Nano Research, 2022: 1-13. Biomolecules, 2022, 12(5): 636. Biomater. Sci., 2022, DOI: 10.1039/D2BM00719C.

Reviewer 2 Report

The article from Corina Popovici and coworkers investigate the encapsulation of a synthetic compound into Pluronic F127 micelles for the treatment of bacterial oral diseases. They have widely characterized the micelles, study their stability, hemolysis degree, in vitro cytotoxicity as well as the in vitro antimicrobial activity.  Although the article is very interesting and well-structured, the abstract must be improved explaining clearly what is the unmeet clinical need that you want to treat (I think is the treatment of bacterial oral diseases), the way of administration (by injection or topical application) and that the API is  [1-(5′-nitrobenzimidazole-2′-yl-sulphonyl-ace-98 tyl)-4-aryl-thiosemicarbazide and it was synthesized and fully characterized in your previous publication . Moreover, the authors can find attached few minor suggestions.
